# REVISITING DEEP AUDIO-TEXT RETRIEVAL THROUGH THE LENS OF TRANSPORTATION

**Manh Luong[1], Khai Nguyen[2], Nhat Ho[2], Dinh Phung[1], Gholamreza Haffari[1], Lizhen Qu[1]**
[1]Monash University, Australia, [2] University of Texas at Austin, USA
`{tien.luong,dinh.phung,gholamreza.haffari,lizhen.qu}@monash.edu`
`{khainb,minhnhat}@utexas.edu`

## ABSTRACT

The Learning-to-match (LTM) framework proves to be an effective inverse optimal transport approach for learning the underlying ground metric between two sources of data, facilitating subsequent matching. However, the conventional LTM framework faces scalability challenges, necessitating the use of the entire dataset each time the parameters of the ground metric are updated. In adapting LTM to the deep learning context, we introduce the mini-batch Learning-to-match (m-LTM) framework for audio-text retrieval problems. This framework leverages mini-batch subsampling and Mahalanobis-enhanced family of ground metrics. Moreover, to cope with misaligned training data in practice, we propose a variant using partial optimal transport to mitigate the harm of misaligned data pairs in training data. We conduct extensive experiments on audio-text matching problems using three datasets: AudioCaps, Clotho, and ESC-50. Results demonstrate that our proposed method is capable of learning rich and expressive joint embedding space, which achieves SOTA performance. Beyond this, the proposed m-LTM framework is able to close the modality gap across audio and text embedding, which surpasses both triplet and contrastive loss in the zero-shot sound event detection task on the ESC-50 dataset. Notably, our strategy of employing partial optimal transport with m-LTM demonstrates greater noise tolerance than contrastive loss, especially under varying noise ratios in training data on the AudioCaps dataset. Our code is available at https://github.com/v-manhlt3/m-LTM-Audio-Text-Retrieval

## 1 INTRODUCTION

Audio-text matching is a crucial task that has tremendous applications, such as audio-text searching (Elizalde et al., 2019; Koepke et al., 2022), audio captioning (Kim et al., 2019), and text-to-audio generation (Liu et al., 2023). The mainstream approaches for audio-text matching involve the construction of a shared embedding space that encompasses both audio and text modalities. The shared embedding space should ideally be endowed with richness and expressiveness. The primary objective of learning a joint audio-text embedding space is to establish a seamless alignment between all text captions and their corresponding audio counterparts. Moreover, it is crucial that this embedding space should minimize the modality gap between audio and text embeddings, thereby enhancing its transferability to downstream tasks (Liang et al., 2022).

The widely adopted approaches for learning the joint embedding space are contrastive learning (Jia et al., 2021; Radford et al., 2021; Mei et al., 2022; Wu et al., 2022b) , and triplet loss (Wei et al., 2021). Both methods have shown their success in learning the joint embedding space cross modalities. Even though they are effective in training cross-modal matching models, these methods only aim to minimize the ground metric, either Euclidean distance or cosine similarity, for an anchor point at a time and push away all negative samples. They treat all negative samples equally, therefore, they might ignore the geometric properties of the joint embedding space. As a consequence, a suboptimal metric may make the relative distances between a set of positively aligned embeddings similar to those between misaligned embeddings. Furthermore, (Liang et al., 2022) showed that contrastive learning preserves the modality gap in the joint embedding space. Therefore, contrastive learning hinders the transferability of the joint embedding space to downstream tasks.

An additional hurdle encountered in the process of acquiring a joint embedding space across modalities is the presence of noisy data, which is prevalent in audio. The noisy data for multimodal matching is misaligned data pairs due to the data collection process. It is desirable that the examples in a training dataset for multimodal matching are precisely aligned across modalities. If the multimodal training data is collected from the Internet (Jia et al., 2021; Sharma et al., 2018), it is unavoidable to include substantial noisy correspondence pairs as ground truth. To address this issue, (Huang et al., 2021) introduces the Noise Correspondence Rectifier approach for image-text matching. However, this technique heavily depends on a particular choice of neural network architectures, therefore, it is less amenable to straightforward adaptation with alternative tasks.

To deal with the aforementioned issues, we propose the minibatch Learning-to-Match (m-LTM) framework to learn the joint embedding space across audio and text modalities through the lens of optimal transport. Specifically, we formulate the cross-modal matching problem as solving an optimal transport problem with entropic regularization in the embedding space to match the point sets between audios and texts. We derive a scalable solution to the problem based on mini-batches to ensure that it scales up to large audio-text datasets. The resulting optimal plan is an implicit ranking matrix in which each row points out the best match between two modalities and ranks all negative points from one modality to the other. To further reduce the modality gap between audio and text embeddings, we replace the widely used Euclidean distance with Mahalanobis distance (Dhouib et al., 2020) as the ground metric, which is introduced for the first time for audio-text matching. To mitigate the prevalent alignment noises in the training data for the first time for audio-text matching, we address the problem by using Partial Optimal Transport (POT) (Chapel & Alaya, 2020). To sum up, our key contributions are three-fold:

1. We adapt the learning-to-match framework (Li et al., 2018) to the deep learning setting based on a subsampling training procedure for audio-text matching, which leads to the mini-batch learning-to-match framework. We leverage the soft-matching of the entropic optimal plan and Mahalanobis distance as a ground metric to learn a shared embedding space. Our approach facilitates the acquisition of a shared embedding space that is rich and expressive, thus we acquire SOTA performance in audio-text retrieval tasks on AudioCaps and Clotho datasets.

2. The mini-batch m-LTM framework is capable of bridging the modality gap, therefore, it is more transferable to downstream tasks than triplet and contrastive learning. We evaluate the modality gap across audio and text embedding on three datasets: AudioCaps, Clotho, and ESC50. We also carry out zeroshot sound event detection on ESC50 dataset to illustrate the transferability of the m-LTM framework.

3. We propose a variant of the m-LTM framework which utilizes Partial Optimal Transport (POT) to deal with noisy training data circumstances. Our experiments demonstrate that the minibatch LTM with POT is more noise-tolerance than triplet and contrastive loss on AudioCaps dataset.

## 2 PRELIMINARY

### 2.1 DEEP AUDIO-TEXT RETRIEVAL

Audio-text retrieval is to learn the cross-modality alignment between audio and text captions. The alignment can be achieved by learning a joint embedding space across audio and text modalities. Recently, contrastive learning has been the most effective learning method for learning expressive cross-modal embedding space (Jia et al., 2021; Wu et al., 2022b; Radford et al., 2021).

**Training**. Denote $D = \{(x_i, y_i)\}_{i=1}^{n}$ as all pairs of audio and caption in the training dataset. The goal is to learn a joint embedding space between audio and text embedding. Given an audio encoder $f_\theta(.)$ and a text encoder $g_\phi(.)$, the objective function for learning the embedding space is:

$$
\min_{\theta, \phi} \mathbb{E}_{(X^b, Y^b) \sim D} \left[ -\sum_{i=1}^{b} \log \left( \frac{\exp(s(f_\theta(x_i), g_\phi(y_i))/\tau)}{\sum_{k=1}^{b} \exp(s(f_\theta(x_i), g_\phi(y_k))/\tau)} \right) \right.
$$
$$
\left. -\sum_{i=1}^{b} \log \left( \frac{\exp(s(f_\theta(x_i), g_\phi(y_i))/\tau)}{\sum_{k=1}^{b} \exp(s(f_\theta(x_k), g_\phi(y_i))/\tau)} \right) \right]
\tag{1}
$$

, where $(X^b, Y^b)$ is a minibatch of $b$ audio-caption pairs drawn from training data $D$, $\tau$ is the temperature hyperparameter, and $s(.,.)$ is the cosine similarity between audio and text embedding

**Retrieval**. Given $n'$ audio $X_{test} = \{x_i\}_{i=1}^{n'}$ and $m'$ caption $Y_{test} = \{y_j\}_{j=1}^{m'}$, the ranking matrix $R(X_{test}, Y_{test})$ between two sets of audio and caption can be retrieved by computing pairwise cosine similarity between audio and text embedding. For example, the correspondence caption of a given audio $x_i$ is

$$\hat{y} = \underset{y_j \in Y_{test}}{\arg\max} \frac{\exp(s(f_\theta(x_i), g_\phi(y_j)))}{\sum_{k=1}^{m'} \exp(s(f_\theta(x_i), g_\phi(y_k)))} \tag{2}$$

## 2.2 LEARNING TO MATCH

**Entropic optimal transport.** Given two empirical probability measures $P_X = \frac{1}{n} \sum_{i=1}^n \delta_{x_i}$ and $P_Y = \frac{1}{m} \sum_{j=1}^m \delta_{y_j}$ where $x_i \in \mathcal{X}$ for $i = 1, \ldots, n$ and $y_j \in \mathcal{Y}$ for $j = 1, \ldots, m$, the entropic optimal transport between $P_X$ and $P_Y$ (Cuturi, 2013) with the ground metric $c : \mathcal{X} \times \mathcal{Y} \to \mathbb{R}^+$ is defined as:

$$\pi_{\epsilon,c}^{X,Y} = \underset{\pi \in \Pi(P_X, P_Y)}{\arg\min} \sum_{i=1}^n \sum_{j=1}^n \pi_{ij} c(x_i, y_j) - \epsilon \sum_{i=1}^n \sum_{j=1}^m \pi_{ij} \log \pi_{ij}, \tag{3}$$

where $\Pi(P_X, P_Y) = \{\pi \in \mathbb{R}^{n \times m} | \pi \mathbf{1}_m = \mathbf{1}_n/n, \pi^T \mathbf{1}_n = \mathbf{1}_m/m\}$ denotes the set of transportation plans or couplings between $P_X$ and $P_Y$. When $\epsilon > 0$, the optimization problem Equation 3 can be solved by using the differentiable Sinkorn algorithm (Marshall & Olkin, 1968).

**Learning to match.** In practice, we are given $\{(x_i, y_i)\}_{i=1}^n$ which is all pairs of audio and caption. Therefore, we would like to infer the underlying ground cost metric $c_\phi(x_i, y_j)$ (parameterized by $\phi$). Learning-to-match (LTM) (Li et al., 2019) is one of the most effective solutions for doing such inference. The LTM framework considers the following optimization:

$$\inf_{c \in \mathcal{C}} \text{KL}(\hat{\pi} || \pi_{\epsilon,c}^{X,Y}), \tag{4}$$

where $\mathcal{C}$ is a set of ground metrics, $X$ denotes the set of $\{x_1, \ldots, x_n\}$ and $Y$ denotes the set of $\{y_1, \ldots, y_n\}$, and $\hat{\pi}$ is a matrix of size $n \times n$ such that $\hat{\pi}_{ii} = 1/n$ and $\hat{\pi}_{ij} = 0$ $(i \neq j)$ for $i = 1, \ldots, n$ and $j = 1, \ldots, n$.

**Retrieval.** Given $n'$ audio $X_{test} = \{x_i\}_{i=1}^{n'}$, $m'$ caption $Y_{test} = \{y_j\}_{j=1}^{m'}$, and the optimized ground metric $c$ from the Equation. 4, the ranking matrix $R(X_{test}, Y_{test})$ between two sets of audio and text caption is the optimal solution of the Equation. 3 between $P_{X_{test}} = \frac{1}{n'} \sum_{i=1}^{n'} \delta_{x_i}$, where $x_i \in X_{test}$ and $P_{Y_{test}} = \frac{1}{m'} \sum_{j=1}^{m'} \delta_{y_j}$, where $y_j \in Y_{test}$. For example, the correspondence caption of a given audio $x_i$ is

$$\hat{y} = \underset{y_j \in Y_{test}}{\arg\max} \pi_{\epsilon,c}^{x_i, y_j} \tag{5}$$

, where $\pi_{\epsilon,c}^{x_i, y_j}$ is the coupling value of *i-th* audio and *j-th* text from the optimal plan. The same retrieval procedure can be done for text-to-audio retrieval.

# 3 DEEP AUDIO-TEXT MATCHING VIA MINI-BATCH LEARNING TO MATCH

This section introduces our proposed method, the m-LTM framework, for the audio-text matching task and a variant to deal with noisy correspondence training data. Our model consists of the audio encoder $f_\theta(.)$, the text encoder $g_\phi(.)$, and the interaction matrix $M$. Our proposed framework utilizes the soft-matching from entropic optimal transport and incorporates the Mahanalobis distance as a ground metric to learn a flexible ground metric to solve audio-text matching problems.

## 3.1 MINI-BATCH LEARNING TO MATCH

By abuse of notation, we denote $(X^b, Y^b) \sim D$ as sampling uniformly a mini-batch which contains $b$ pairs of audio and caption. The mini-batch learning-to-match framework is defined as follows:

**Definition 1.** *Given a paired training data $D = \{(x_i, y_i)\}_{i=1}^n$, the mini-batch learning-to-match (m-LTM) framework solves the following optimization problem:*

$$\inf_{c \in \mathcal{C}} \mathbb{E}_{(X^b, Y^b) \sim D}[KL(\hat{\pi}^b || \pi_{\epsilon,c}^{X^b, Y^b})], \tag{6}$$

*where $b \geq 2$ is the mini-batch size, $\pi_{\epsilon,c}^{X^b, Y^b}$ is the optimal transportation plan from Equation 3 with the corresponding empirical measures over mini-batches $X^b$ and $Y^b$, and $\hat{\pi}_b$ is a matrix of size $b \times b$ such that $\hat{\pi}_{ii} = 1/b$ and $\hat{\pi}_{ij} = 0$ ($i \neq j$) for $i = 1, \ldots, b$ and $j = 1, \ldots, b$.*

**Neural network parameterization.** As in the preliminary section, we parameterize the ground metric $c_{\theta,\phi}(x, y) = d(f_\theta(x_i), g_\phi(y_j)))$ where $f_\theta : \mathcal{X} \to \mathcal{Z}$ ($\theta \in \Theta$) and $g_\phi \to \mathcal{Z}$ ($\phi \in \Phi$) are two neural networks parameterized by $\theta$ and $\phi$ respectively, and $d : \mathcal{Z} \times \mathcal{Z} \to \mathbb{R}^+$ is a chosen metric e.g., $\mathbb{L}_2$ norm. Therefore, the set of ground metric $\mathcal{C}$ becomes the set of all possible weights of two neural networks i.e., $\Theta \times \phi$. As a result, Equation 6 is equivalent to:

$$\min_{(\theta,\phi) \in \Theta \times \Phi} \mathbb{E}_{(X^b, Y^b) \sim D}[\text{KL}(\hat{\pi}^b || \pi_{\epsilon,c_{\theta,\phi}}^{X^b, Y^b})]. \tag{7}$$

**Stochastic gradient estimation.** To solve the optimization in Equation 7, we perform stochastic gradient descent algorithm to update $\theta$ and $\phi$. In particular, we sample $B \geq 1$ mini-batches $(X^b, Y^b)_1, \ldots, (X^b, Y^b)_B \overset{i.i.d}{\sim} D$, then form the following stochastic estimation of gradients:

$$\nabla_{(\theta,\phi)} \mathbb{E}_{(X^b, Y^b) \sim D}[\text{KL}(\hat{\pi}^b || \pi_{\epsilon,c_{\theta,\phi}}^{X^b, Y^b})] \approx \frac{1}{B} \sum_{i=1}^{B} \nabla_{(\theta,\phi)} \text{KL}(\hat{\pi}^b || \pi_{\epsilon,c_{\theta,\phi}}^{(X^b, Y^b)_i})]. \tag{8}$$

Since this estimation is unbiased, the stochastic optimization procedure can converge to a local minima. In practice, we often set $B = 1$ for having a computationally fast estimation.

**Retrieval.** During the inference phase, it is noteworthy that the quantity of audio and caption data is considerably smaller compared to the extensive training dataset. Moreover, the conventional evaluation protocol for retrieval necessitates that each query must be compared to the entirety of available items in the test set to compute the evaluation metric. Consequently, employing the minibatch configuration during the retrieval phase does not yield discernible advantages. For more detailed information on retrieval, please refer to Section 2.2.

## 3.2 MAHALANOBIS-ENHANCED GROUND METRIC

Although the parameterization of the ground metric in Section 3.1 i.e., $c_{\theta,\phi}(x, y) = d(f_\theta(x_i), g_\phi(y_j)))$ has been widely used in practice, it still has a limitation. In particular, the embedding from the two encoders $f_\theta(.)$ and $g_\phi(.)$ might not be well aligned in terms of scaling in each dimension. To partially address the issues, we propose a more generalized family of ground metric which is based on the Mahalanobis distance.

**Definition 2.** *Given two encoder functions $f_\theta : \mathcal{X} \to \mathcal{Z}$ and $g_\phi : \mathcal{Y} \to \mathcal{Z}$, a metric $d : \mathcal{Z} \times \mathcal{Z} \to \mathbb{R}^+$, the Mahalanobis enhanced ground metric is defined as:*

$$c_{\theta,\phi,M}(x, y) = \sqrt{(f_\theta(x_i) - g_\phi(y_j))^\top M (f_\theta(x_i) - g_\phi(y_j))}, \tag{9}$$

*for $\theta \in \Theta$ and $\phi \in \Phi$ which are spaces of parameters and $M$ is a positive definite matrix.*

**Mini-batch learning to match with Mahalanobis-Enhanced Ground Metric.** By using the family of Mahalanobis-Enhanced ground metrics in Definition 2, the m-LTM objective in Definition 1 becomes:

$$\min_{(\theta,\phi,M) \in \Theta \times \Phi \times \mathcal{M}} \mathbb{E}_{(X^b, Y^b) \sim D}[\text{KL}(\hat{\pi}^b || \pi_{\epsilon,c_{\theta,\phi,M}}^{X^b, Y^b})], \tag{10}$$

where $\mathcal{M}$ is the set of all possible positive definite matrices e.g., $x^\top M x > 0$ for all $x \in \mathcal{Z}$.

**Hybrid stochastic gradient descent.** the optimization problem in Equation 10 consists of three parameters $\theta, \phi$, and $M$. In contrast to $\theta$ and $\phi$ which are unconstrained, $M$ is a constrained parameter. Therefore, we propose to use a hybrid stochastic gradient descent algorithm. In particular, we still

update $\theta, \phi$ using the estimated gradients in Equation 8. However, we update $M$ using the projected gradient descent update (Rockafellar, 1997). We first estimate the stochastic gradient with respect to $M$:

$$\nabla_M \mathbb{E}_{(X^b, Y^b) \sim D}[\text{KL}(\hat{\pi}^b || \pi_{\epsilon, c_{\theta, \phi}, M}^{X^b, Y^b})] \approx \frac{1}{B} \sum_{i=1}^{B} \nabla_M \text{KL}(\hat{\pi}^b || \pi_{\epsilon, c_{\theta, \phi}, M}^{(X^b, Y^b)_i}). \tag{11}$$

After that, we update $M = \text{Proj}(F(M, \nabla M))$ where $F(M, \nabla M)$ denotes the one-step update from a chosen optimization scheme e.g., Adam, gradient descent, and so on, and $\text{Proj}(\cdot)$ denotes the projecting function that maps a unconstrained matrix to the space of positive definite matrix $\mathcal{M}$. In greater detail, $\text{Proj}(A) = U \text{diag}(\bar{S}) V^T$ with $U, S, V = SVD(A)$ and $\bar{S} = \text{diag}(\max\{0, \sigma_1\}, ..., \max\{0, \sigma_n\})$. We refer the reader to the training algorithm in the Algorithm. 1 in the Appendix A.2.

### 3.3 PARTIAL OT FOR NOISY CORRESPONDENCE

**Noisy correspondence setting.** We first introduce the noisy correspondence setting for the audio-text matching task. That is, given the training data $D = \{(x_i, y_i)\}_{i=1}^N$ where $N$ is the number of training samples, a proportion of training data $N_{cor}$, $N_{cor} < N$, is corrupted, for instance, due to the data collection process. The amount of data corruption is unknown, and we do not know which samples are corrupt, thus presenting a challenging learning scenario. Due to the noisy correspondence phenomenon, the empirical matching $\hat{\pi}$ is now corrupted to an incomplete matching $\bar{\pi}$, in which there is no matching in some rows. Assuming an absence of noise in the original training data, we introduce a noisy correspondence training dataset by randomly shuffling a portion of audio-text pairs within the dataset. We denote a random variable $z \in \{0, 1\}$ which is sampled from a binomial distribution $Binomial(N, \frac{N_{cor}}{N})$, if $z = 1$ indicates the audio-text pair is shuffled. The training data is now $\tilde{D} = \{(z_i, x_i, y_i)\}_{i=1}^N$, where $z_i \sim Binomial(N, \frac{N_{cor}}{N})$.

Given the absence of knowledge regarding the nature of data corruption, the training process relies solely on the empirical matching $\hat{\pi}$. The training objective is to infer the incomplete matching $\bar{\pi}$ from the noisy training dataset $\tilde{D}$. However, it is important to note that the m-LTM framework may encounter challenges when attempting to recover the incomplete matching $\bar{\pi}$. This difficulty arises due to the constraint imposed by transportation preservation, which dictates that each row must be associated with at least one column.

To solve the aforementioned issue, we propose to use Partial OT, which relaxes the transportation preservation constraint, to mitigate the harmfulness of noisy empirical matching for approximating the incomplete matching $\bar{\pi}$. The objective function 10 is rewritten as

$$\min_{(\theta, \gamma, M) \in \Theta \times \Phi \times \mathcal{M}} \mathbb{E}_{(\tilde{X}^b, \tilde{Y}^b) \sim \tilde{D}}[\text{KL}(\hat{\pi}^b || \pi_{s, \epsilon, c_{\theta, \phi}, M}^{\tilde{X}^b, \tilde{Y}^b})],, \tag{12}$$

, where $(\tilde{X}^b, \tilde{Y}^b)$ is a minibatch sampled from noisy training data $\tilde{D}$, and $\pi_{\epsilon, s}^{\tilde{X}^b, \tilde{Y}^b}$ is the optimal solution of the equation

$$\pi_{s, \epsilon, c_{\theta, \gamma}, M}^{\tilde{X}^b, \tilde{Y}^b} = \arg\min_{\pi \in \Pi_s(P_{\tilde{X}^b}, P_{\tilde{Y}^b})} \sum_{i=1}^{b} \sum_{j=1}^{b} \pi_{ij} c(x_i, y_j) - \epsilon \sum_{i=1}^{b} \sum_{j=1}^{b} \pi_{ij} \log \pi_{ij}, \tag{13}$$

where $\Pi_s(P_{\tilde{X}^b}, P_{\tilde{Y}^b}) = \{\pi \in \mathbb{R}_+^{b \times b} | \pi \mathbb{1} \leq P_{\tilde{X}^b}, \pi^\top \mathbb{1} \leq P_{\tilde{Y}^b}, \mathbb{1} \pi^\top \mathbb{1} = s\}$. The partial OT optimization can be solved by using Bregman projection (Benamou et al., 2015). Intuitively, the partial transportation plan should be such that true correspondence audio-text pairs within a minibatch are mapped to each other, while noisy correspondence audio-text pairs are discarded from the transportation plan. We conducted an experiment to examine the relationship between the transportation mass and noise ratio in training data in the Section. 5.2

## 4 RELATED WORKS

**Cross-modal Matching.** Cross-modal matching is to learn a shared representation space (Wang et al., 2013) between two modalities like image-text (Jia et al., 2021; Radford et al., 2021; Wei et al.,

2020) or audio-text (Wu et al., 2022b; Deshmukh et al., 2022; Oncescu et al., 2021; Mei et al., 2022). The most popular approach for cross-modal matching is metric learning, such as triplet loss (Wei et al., 2020) and contrastive loss (Radford et al., 2021; Yang et al., 2022). The cross-modal training procedure exposes a great benefit of the transferability to downstream tasks (Zeng et al., 2022) . (Xu et al., 2021) shows that contrastive cross-modal pretraining procedure can be used to train a single model for both text and video understanding tasks without fine-tuning. However, cross-modal training is beneficial for downstream tasks, this training procedure requires a huge amount of data with high alignment quality. (Huang et al., 2021) introduces noisy correspondence training circumstances for image-text matching. This work demonstrates that their proposed method is capable of dealing with high-level image-caption mismatching, but it heavily depends on neural network architecture. To handle the noisy correspondence training data, we introduce a new loss function that is not only capable of mitigating the harmfulness of label mismatch for cross-modal matching but also easy to incorporate into existing backbones for cross-modal matching. Furthermore, we apply a new ground metric, Mahalanobis distance, which is able to learn a rich and expressive joint embedding space across audio-text modalities and close the gap between audio and text embedding space.

**Optimal transport**. Optimal transport (OT) is a powerful mathematical tool to measure the discrepancy between two probability distributions. Therefore, it has a wide range of applications, such as domain adaption (Courty et al., 2017), generative models (Genevay et al., 2018; Nguyen & Ho, 2024; Nguyen et al., 2024), and representation learning (Chen et al., 2020). Our work is close to the Inverse Optimal Transport (IOT) problem, which has been well-studied (Dupuy et al., 2016; Li et al., 2018; Stuart & Wolfram, 2019; Chiu et al., 2022) by the research community aiming to learn matching between two sources of interest. (Wu et al., 2022a) proposed the OTTER framework which leverages entropic optimal plan as soft matching labels to learn many-to-many relationships in the image-text matching task based on contrastive learning procedure. (Shi et al., 2023) proposed a unifying understanding for contrastive learning from an inverse optimal transport view. All previous works either adapt cosine similarity as the ground metric (Shi et al., 2023) or study the whole empirical plan of tabular data for learning-to-matching (Dupuy et al., 2016; Li et al., 2018; Stuart & Wolfram, 2019). Mini-batch optimal transport has been investigated in Fatras et al. (2020); Nguyen et al. (2022b;a). In our paper, we adapt the learning-to-match framework (Li et al., 2018) for the minibatch setting with a more flexible ground cost metric, Mahalanobis distance, to model the expressive shared embedding space between audio and text modalities. Mahalanobis distance represents an infinite set of cost functions in the theoretical analysis of robust optimal transport optimization (Paty & Cuturi, 2019; Dhouib et al., 2020). However, there is a gap between theory and application of Mahalanobis distance as a ground metric for optimal transport optimization due to the difficult constraint of Mahalanobis distance. We close the gap by successfully applying Mahalanobis distance for the audio-text matching application. The strict constraint for learning Mahalanobis distance is relaxed by Projection Gradient Descent.

## 5 EXPERIMENTS

In this section, we design experiments to examine the effectiveness of our framework in order to learn a joint embedding space across audio and text modalities. We conduct the audio-text retrieval task on two datasets: AudioCaps (Kim et al., 2019) and Clotho (Drossos et al., 2019) to answer the question of whether our framework is able to learn a rich and expressive joint embedding space. In addition, we evaluate the modality gap metric (Liang et al., 2022) in the joint embedding space on three datasets: AudioCaps, Clotho, and ESC-50 (Piczak, 2015) to illustrate that the m-LTM framework is capable of bridging the modality gap between audio and text embedding space. We perform zero-short sound event detection on ESC-50 test set to demonstrate the transferability of our method. We verify the noise-tolerance capability of the m-LTM with POT framework in a variant of noise ratio training data on the AudioCaps dataset. Finally, we conduct ablation studies for hyperparameters tuning. The implementation details of our framework are provided in Appendix. A.1

**Baselines**. We compare against all state-of-the-art models (Mei et al., 2022; Deshmukh et al., 2022; Oncescu et al., 2021; Wu et al., 2022b) for audio-text retrieval tasks to illustrate the remarkable performance of our framework. All the results for these baselines are copied from the reference papers. Furthermore, we examine our framework with diverse learning objectives, such as contrastive and triplet loss.

**Table 1:** The comparison of m-LTM framework with baselines on audio-text retrieval task on two benchmark datasets, AudioCaps and Clotho dataset.

| Dataset | Method | Text->Audio | | | Audio->Text | | |
|---|---|---|---|---|---|---|---|
| | | R@1 | R@5 | R@10 | R@1 | R@5 | R@10 |
| Audiocaps | (Oncescu et al., 2021) | 28.1 | - | 79.0 | 33.7 | - | 83.7 |
| | (Mei et al., 2022) | 33.9 | 69.7 | 82.6 | 39.4 | 72 | 83.9 |
| | (Deshmukh et al., 2022) | 33.07 | 67.30 | 80.3 | 39.76 | 73.72 | 84.64 |
| | (Wu et al., 2022b) | 36.7 | 70.9 | 83.2 | 45.3 | 78 | 87.7 |
| | m-LTM(our) | **39.10** | **74.06** | **85.78** | **49.94** | **80.77** | **90.49** |
| Clotho | (Oncescu et al., 2021) | 9.6 | - | 40.1 | 10.7 | - | 40.8 |
| | (Mei et al., 2022) | 14.4 | 36.6 | 49.9 | 16.2 | 37.5 | 50.2 |
| | (Deshmukh et al., 2022) | 15.79 | 36.78 | 49.93 | 17.42 | 40.57 | 54.26 |
| | (Wu et al., 2022b) | 12.0 | 31.6 | 43.9 | 15.7 | 36.9 | 51.3 |
| | m-LTM(our) | **16.65** | **39.78** | **52.84** | **22.1** | **44.4** | **56.74** |

**Table 2:** Experiments of loss functions on the AudioCaps dataset. There are two audio encoders, ResNet38 and HTSAT, used to extract audio embedding, and the text encoder is the BERT text encoder.

| | Loss | Text->Audio | | | Audio->Text | | |
|---|---|---|---|---|---|---|---|
| | | R@1 | R@5 | R@10 | R@1 | R@5 | R@10 |
| ResNet38 Audio Encoder | Triplet | 32.2 | 68.2 | 81.6 | 36.1 | 69.2 | 81.4 |
| | Contrastive | 33.9 | 69.7 | 82.6 | 39.4 | 72.0 | 83.9 |
| | m-LTM(our) | **39.10** | **74.06** | **85.78** | **49.94** | **80.77** | **90.49** |
| HTSAT Audio Encoder | Triplet | 33.87 | 69.91 | 83.23 | 33.95 | 70.03 | 82.46 |
| | Contrastive | 34.38 | 71.16 | 84.64 | 35.84 | 71.06 | 83.39 |
| | m-LTM(our) | **40.02** | **75.86** | **86.42** | **40.13** | **74.09** | **85.89** |

**Evaluation metrics**. To evaluate our proposed framework, we use the recall at rank $k (R@k)$ metric, which is the standard evaluation metric for cross-modal retrieval tasks. $R@k$ reports the percentage of the ground-truth retrieval within the top-k ranks, thus, higher is better. $R@1$, $R@5$, and $R@10$ are shown to compare performance among baselines and our method. We also report the modality gap metric (Liang et al., 2022) to examine the shared embedding space. The lower modality gap is better for downstream tasks.

## 5.1 EXPRESSIVENESS AND TRANSFERABILITY OF THE JOINT EMBEDDING SPACE

We compare our proposed framework with state-of-the-art models in the audio-caption retrieval task to verify the expressiveness of our proposed framework. To have a fair comparison, we train all the models and objective functions with the same amount of training data. As shown in Table. 1, our minibatch learning-to-match framework outperforms four baseline methods in terms of $R@1$, $R@5$, and $R@10$ evaluation metrics on two datasets in two retrieval tasks, audio-to-text and text-to-audio retrievals. Furthermore, we carry out the experiment to compare the m-LTM objective function with contrastive and triplet loss on the AudioCaps dataset shown in Table. 2. Our method illustrates the highest performance for cross-modal retrieval tasks, there is a significant gap regarding the $R@1$ score compared with the best loss function, contrastive loss. For the ResNet audio encoder, the utilization of m-LTM results in a marked and notable augmentation of the $R@1$ score, showcasing an improvement from $33.9\%$ to $37.97\%$ for the text-to-audio matching task, and a distinct escalation from $39.4\%$ to $47.43\%$ for the audio-to-text matching task. This performance increase is also discernible in the case of the HTSAT audio encoder. We also provide some qualitative results for audio-text retrieval tasks in Appendix A.3.

**Table 3:** The zero-shot sound event detection on the ESC50 test set, the $R@1$ score is equivalent to accuracy.

| Loss | Audio->Sound | | | |
|---|---|---|---|---|
| | R@1 | R@5 | R@10 | mAP |
| Triplet | 71.25 | 91.75 | 95.75 | 80.09 |
| Contrastive | 72.25 | 93 | 96.75 | 80.84 |
| m-LTM | 81.0 | 97.0 | 99.25 | 87.57 |

**Table 4:** The modality gap between audio and text embedding in the shared embedding space. Lower is better for downstream tasks.

| Loss | Modality gap($\|\vec{\Delta}_{gap}\|$) | | |
|---|---|---|---|
| | AudioCaps | Clotho | ESC50 |
| Triplet | 0.149 | 0.283 | 0.937 |
| Contrastive | 0.181 | 0.266 | 0.922 |
| m-LTM | 0.117 | 0.142 | 0.224 |

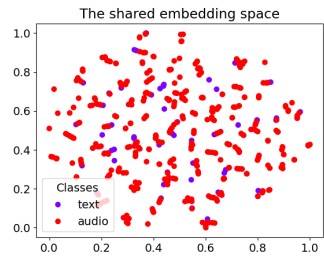
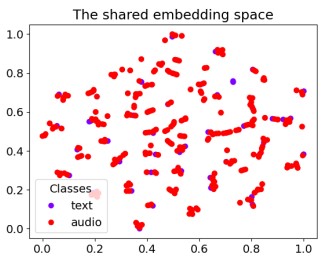

(a) Contrastive Learning    (b) Minibatch learning-to-match

**Figure 1:** The visualization of the shared embedding space between audio and text embedding on the ESC50 test set based on tSNE algorithm. Each text embedding represents a label in the test set, and each audio embedding represents an audio in the test set.

**Table 5:** The performance of learning-to-match and metric learning methods for audio-text retrieval task under the variant ratio of noisy training data.

| Noise | Method | Text->Audio | | | Audio->Text | | |
|---|---|---|---|---|---|---|---|
| | | R@1 | R@5 | R@10 | R@1 | R@5 | R@10 |
| 20% | Triplet loss | 23.01 | 54.98 | 69.98 | 28.52 | 58.09 | 70.11 |
| | Contrastive loss | 31.34 | 67.73 | 81.27 | 40.12 | 70.84 | 82.54 |
| | m-LTM | 35.51 | 71.32 | 84.01 | 46.64 | 78.68 | 87.87 |
| | m-LTM with POT | **35.92** | **72.28** | **84.11** | **47.12** | **79.2** | **88.19** |
| 40% | Triplet loss | 0.1 | 1.19 | 2.75 | 1.25 | 5.43 | 9.4 |
| | Contrastive loss | 26.68 | 62.98 | 78.18 | 34.69 | 66.66 | 78.99 |
| | m-LTM | 32.58 | 67.75 | 80.89 | 40.31 | 71.16 | 84.57 |
| | m-LTM with POT | **33.64** | **69.23** | **82.27** | **42.63** | **73.35** | **86.1** |
| 60% | Triplet loss | 0.1 | 0.52 | 1.06 | 0.1 | 0.52 | 1.46 |
| | Contrastive loss | 20.58 | 53.96 | 70.72 | 27.37 | 58.72 | 75.21 |
| | m-LTM | 25.26 | 59.72 | 75.03 | 34.08 | 66.77 | 79.62 |
| | m-LTM with POT | **27.73** | **62.61** | **76.17** | **35.42** | **68.65** | **80.56** |

We conduct a zero-shot sound event detection experiment on the ESC50 test set to examine the transferability of the learned embedding space. We replace the task of sound event detection with audio-text retrieval, all the classes in the test set are converted to the template caption as "This is a sound of {class}". We pretrain all models on the AudioCaps dataset on the audio-caption matching task. We report $R@1$, $R@5$, $R@10$, and mAP scores for triplet loss, contrastive loss, and the m-LTM loss in Table. 3. The m-LTM loss acquires the highest $R@1$ score, equivalent to achieving the highest accuracy for the sound event detection task. Also, there is a large performance gap in terms of $R@5$, $R@10$, and mAP metrics between m-LTM loss and contrastive loss. As depicted in Figure. 1, audio embeddings learned from the m-LTM loss are clearly clustered by the text sound event classes, which means the learned embedding space is more expressive for the zero-shot sound event detection setting. In contrast, audio embeddings learned from contrastive loss are spread uniformly in the joint embedding space. This observation consolidates to the result in Table. 3 which is the m-LTM is more transferable than contrastive loss for zero-shot setting. Furthermore, we study the modality gap of our proposed loss in Table. 4, the $\vec{\Delta}_{gap}$ modality gap metric between two modalities is computed as the mean of embedding between two modalities $\vec{\Delta}_{gap} = \frac{1}{n}\sum_{i=1}^{n} f(x_i) - \frac{1}{n}\sum_{i=1}^{n} g(y_i)$. Our proposed loss achieves the smallest gap between audio and text embedding, which means it is more transferable to downstream tasks than triplet and contrastive loss.

## 5.2 NOISY CORRESPONDENCE TOLERANCE

To study the robustness of our proposed framework with noisy correspondence, we carry out the experiment under varying noisy correspondence training data. As defined in the section 3.3, we use three different noise ratios $\xi = \{0.2, 0.4, 0.6\}$ of training data. We perform the experiment on the AudioCaps dataset for four losses: triplet loss, contrastive loss, m-LTM loss, and m-LTM with POT loss described in section 3.3. First, the binary value $z_i$ is sampled from a $Binomial(N, \xi)$ distribution

for each audio-text pair in the training data, so we have N tuples of training data $\{z_i, x_i, y_i\}_{i=1}^N$. If $z_i = 1$ we replace the text $y_i$ with a random text $\tilde{y}_i$ in training data, the noisy training tuple is $\{z_i = 1, x_i, \tilde{y}_i\}$. The random variable $z \sim Binomial(N, \xi)$ is only used for data sampling, and it is not used during training. Table. 5 shows the audio-text retrieval performance under the variant noise ratio of four loss functions: triplet loss, contrastive loss, and our proposed methods. The triplet loss is less robust to high levels of noise, its performance plunges from $28.52\%$ to $1.25\%$ and from $23.01\%$ to $0.1\%$ in terms of $R@1$ score when the noise ratio increases from $20\%$ to $40\%$ for audio-to-text and text-to-audio retrieval, respectively. Contrastive loss and two variants of our proposed method are robust to a wide range of noise ratios, however, our proposed losses surpass contrastive loss for all levels of noise in terms of recall at rank $k$ metrics. There is a significant gap between contrastive loss and the m-LTM with POT loss on the $R@1$ metric for two retrieval tasks. At $60\%$ noise of training data, the m-LTM with POT loss boosts the performance from $20.58\%$ and $27.37\%$ of contrastive loss to $27.73\%$ and $35.42\%$ in terms of $R@1$ metric for text-to-audio and audio-to-text, respectively. Furthermore, it is noteworthy that in both of the two retrieval tasks, the m-LTM with POT consistently outperforms the conventional m-LTM across all levels of noise. As a result, it can be inferred that the m-LTM with POT exhibits state-of-the-art (SOTA) performance in effectively addressing challenging scenarios characterized by noisy correspondences.

## 5.3 ABLATION STUDY

**Table 6:** Ablation studies on AudioCaps dataset for choosing the ground metric and hyper-parameter for minibatch learning-to-match loss function.

| | $\epsilon$ | Text->Audio | | | Audio->Text | | |
|---|---|---|---|---|---|---|---|
| | | R@1 | R@5 | R@10 | R@1 | R@5 | R@10 |
| | 0.05 | 39.10 | 74.06 | 85.78 | 49.94 | 80.77 | 90.49 |
| Maha | 0.1 | 29.86 | 65.51 | 79.64 | 39.81 | 70.53 | 83.07 |
| | 0.5 | 9.59 | 31.99 | 48.19 | 10.55 | 34.16 | 46.91 |
| | 0.05 | 38.13 | 73.47 | 85.43 | 48.15 | 79.10 | 89.23 |
| Eucli | 0.1 | 29.55 | 65.60 | 79.43 | 38.45 | 70.84 | 82.23 |
| | 0.5 | 10.17 | 33.06 | 48.50 | 13.06 | 34.69 | 48.27 |

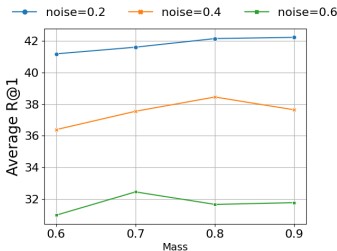

**Figure 2:** Ablation study on the AudioCaps dataset for transportation mass of m-LTM with POT.

In this section, we conduct two ablation studies. As illustrated in Table. 6, Mahalanobis distance as the ground metric for learning-to-match framework surpasses the Euclidean distance for audio-text retrieval task. Also, the $\epsilon$ hyper-parameter is crucial in our framework. If $\epsilon$ is too high, the optimal transportation plan will be a uniform distribution and lead to a decrease in performance. The value of $\epsilon = 0.05$ acquires the highest performance, thus, all the reported results in this paper using $\epsilon = 0.05$. The second ablation study is the effect of transportation mass for noisy correspondence training data demonstrated in Figure. 2, we report the average $R@1$ score for both audio-to-text and text-to-audio retrieval tasks. For different noise ratios, there is a best transportation mass value, which corresponds to noise level, to acquire the highest performance. This observation supports our hypothesis in section. 3.3 which is choosing the appropriate transportation mass for noisy correspondence circumstances is essential.

## 6 CONCLUSION

In this paper, we introduce the mini-batch learning-to-match (m-LTM) framework for audio-text matching, the m-LTM utilizes the Mahalanobis distance and soft-matching from the entropic optimal plan to facilitate learning of a shared embedding space for audio and text modalities. By leveraging soft matching from the optimal plan, our framework is capable of learning a rich and expressive joint embedding space across audio and text modality, therefore, the m-LTM acquires state-of-the-art performance for audio-text retrieval on AudioCaps and Clotho datasets. Furthermore, we adopt the Mahalobis distance as the ground metric to reduce the modality gap, thus, our framework is more transferable to downstream tasks compared to conventional methods relying on triplet and contrastive loss functions. Finally, we propose a variant of the m-LTM with POT to alleviate the adverse effects of noisy correspondence in training data under various noise conditions.

## ACKNOWLEDGEMENTS

NH acknowledges support from the NSF IFML 2019844 and the NSF AI Institute for Foundations of Machine Learning.

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

## A APPENDIX

### A.1 IMPLEMENTATION DETAILS

**AudioCaps** is the biggest audio captioning dataset that consists of around 50k audio-caption pairs. All audio clips are extracted from AudioSet (Gemmeke et al., 2017), a large-scale dataset for audio tagging. There are a total of 40,582 audio clips in training data, and all audio clips are 10 seconds long. Each audio clip has a single human-annotated caption. The validation and test sets have 494 and 957 audio clips, respectively, and each audio clip has five ground-truth captions.

**Clotho** is an audio captioning dataset collected from the Freesound platform. The audio clips' length varies from 15 to 30 seconds. We use the second version of the Clotho dataset to carry out the experiment. There are 3839 audio clips in the training set and over 1,000 audio clips in the validation and test set. All captions in the dataset consist of 5 human-annotated captions.

**ESC50** is an audio dataset for sound event detection. The dataset incorporates 2,000 labeled environmental recordings with 50 classes. We only use the test set of the ESC50 dataset to verify the transferable ability of our proposed method. There are 400 audio clips in the test set used in our experiment.

**Implementation details**. We follow the implementation details in (Mei et al., 2022). Log mel-spectrograms are used as the audio features for the audio encoder, and the settings used to extract Log mel-spectrograms are window size of 1024, Hanning window, hop-size of 320, and 64 first mel bins. The audio encoder is the ResNet-38 model (Kong et al., 2019) that is pretrained on the AudioSet dataset with the audio tagging task. After pertaining, we discard the last two layers and then apply an average-pooling layer to accumulate features along the frequency dimension on the extracted feature map. We also conducted an experiment on the HTSAT audio encoder (Chen et al., 2022), which is a transformer-based encoder. The text encoder is the BERT model (Devlin et al., 2018), which is pretrained on various NLP tasks to be able to extract contextual-aware embeddings. Due to the dimension mismatch between output of the audio encoder and the text encoder, we utilize two linear project blocks to project two encoder outputs to the shared embedding space. The shared embedding space is a 1024-dimensional space. The interaction matrix $M$ is initialized as follows $M = \frac{1}{2}(A + A^T) + \mathbb{1}$, where $A$ is a random initialized matrix of size 1024 by 1024. All the models and the matrix $M$ are trained for 30 epochs with Adam optimizer (Kingma & Ba, 2014). The hyperparameters for training are learning rate $lr = 1 \times 10^{-4}$, batch size $b = 256$, and dropout ratio $p = 0.2$. All experiments are performed on a single A100 GPU.

### A.2 ALGORITHM

---

**Algorithm 1:** Learning ground cost metric using m-LTM framework and Mahanalobis distance

**Input** : Initialize audio encoder $f_\theta$, text encoder $g_\phi$, interaction matrix $M$, and training data $p(x, y)$

**Output :** Learned audio encoder $f_\theta$, text encoder $g_\phi$ and interaction matrix $M$

**while** *until converged* **do**
$\quad (x^i, y^i)_{i=1}^b \sim p(x, y)$;
$\quad Z_1 = \{f_\theta(x^i)\}_{i=1}^b$;
$\quad Z_2 = \{g_\phi(y^i)\}_{i=1}^b$;
$\quad \hat{\pi} = \text{Eye}(\frac{1}{b})$;
$\quad C_M^{Z_1, Z_2} = \sqrt{(Z_1 - Z_2)^T M (Z_1 - Z_2)}$;
$\quad \pi_{\epsilon, c_{\theta, \phi, M}}^{X^b, Y^b} = \text{Sinkhorn}(C_M^{Z_1, Z_2}, \epsilon)$;
$\quad \mathcal{L}(\theta, \phi, M) = KL(\hat{\pi} || \pi_{\epsilon, c_{\theta, \phi, M}}^{X^b, Y^b})$;
$\quad \theta = \theta - \eta \nabla_\theta \mathcal{L}$;
$\quad \phi = \phi - \eta \nabla_\phi \mathcal{L}$;
$\quad M = Proj(M - \eta \nabla_M \mathcal{L})$
**return** $f_\theta, g_\phi,$ *and* $M$

---

**Algorithm 2:** Sinkhorn Algorithm

**Input** : $C \in \mathbb{R}^{n \times n}$ and $\epsilon$

**Output :** $\pi_\epsilon$

$K = \exp(-C/\epsilon), v \leftarrow \frac{1_n}{n}$;
**while** *until converged* **do**
$\quad u = \frac{1_n}{n} \oslash Kv; v = \frac{1_n}{n} \oslash K^T u$;
$\pi_\epsilon = \text{diag}(u) K \text{diag}(v)$;
**return** $\pi_\epsilon$

---

## A.3 QUALITATIVE EXPERIMENTS

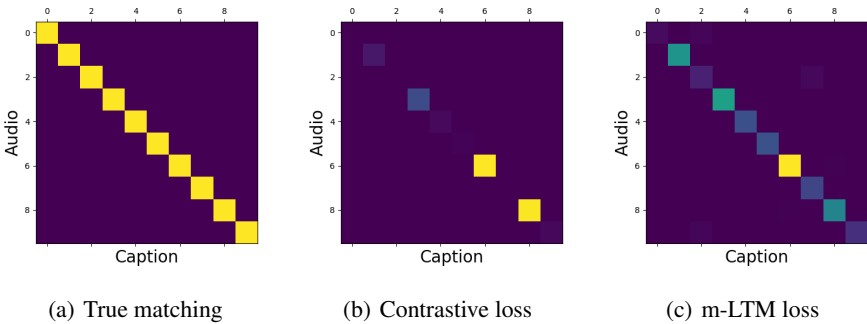

(a) True matching        (b) Contrastive loss        (c) m-LTM loss

**Figure 3:** The visualization of the true matching, and inference matching from pretrained models using contrastive loss and m-LTM loss on ten audio and ten corresponding captions from the test set of AudioCaps dataset.

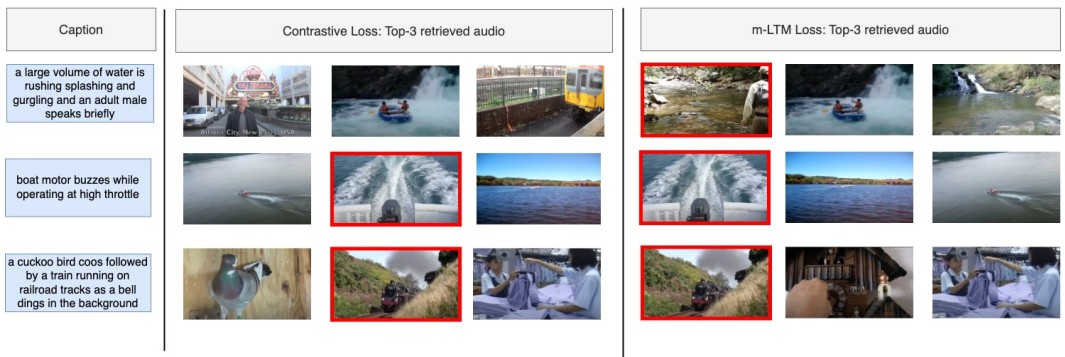

**Figure 4:** Qualitative results for text-to-audio retrieval task. top-1, top-2, and top-3 retrieved audio results are from left to right in the figure. The ground-truth audio for the caption is marked in red border.

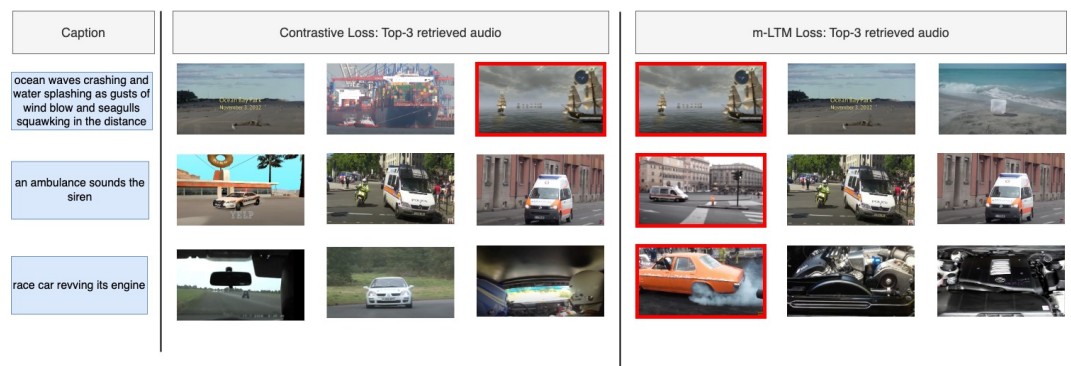

**Figure 5:** Qualitative results for text-to-audio retrieval task. top-1, top-2, and top-3 retrieved audio results are from left to right in the figure. The ground-truth audio for the caption is marked in red border.

## A.4 ADDITIONAL EXPERIMENTS AND ABLATION STUDY

We compare the m-LTM loss with contrastive loss and a variant of contrastive loss with entropic regularization in Table. 7. The training objective of contrastive loss with entropic regularization is as follows

$$\mathcal{L}(\theta, \phi) = \mathcal{L}_{CL} + \epsilon * \mathcal{L}_{ent}(R) \tag{14}$$

**Table 7:** The additional baseline experiment of loss functions, the m-LTM, contrastive loss, and contrastive loss with entropy regularization, on the AudioCaps dataset.

| Audio Encoder | Method | Epsilon | Text->Audio | | | Audio->Text | | |
|---|---|---|---|---|---|---|---|---|
| | | | R@1 | R@5 | R@10 | R@1 | R@5 | R@10 |
| ResNet38 Audio Encoder | CL | - | 33.9 | 69.7 | 82.6 | 39.4 | 72.0 | 83.9 |
| | CL with entropic regularization | 0.05 | 34.04 | 69.11 | 82.51 | 44.11 | 74.71 | 85.37 |
| | | 0.1 | 34.27 | 70.21 | 82.92 | 44.41 | 73.87 | 86.41 |
| | | 0.5 | 33.79 | 69.51 | 82.90 | 42.52 | 72.10 | 85.78 |
| | m-LTM | 0.05 | **39.10** | **74.06** | **85.78** | **49.94** | **80.77** | **90.49** |
| HTSAT Audio Encoder | CL | - | 34.38 | 71.16 | 84.64 | 35.84 | 71.06 | 83.39 |
| | CL with entropic regularization | 0.05 | 35.71 | 72.3 | 85.19 | 37.45 | 73.91 | 84.44 |
| | | 0.1 | 35.12 | 72.05 | 85.01 | 36.96 | 73.55 | 84.34 |
| | | 0.5 | 34.45 | 71.33 | 85.11 | 35.98 | 72.53 | 83.78 |
| | m-LTM | 0.05 | **40.02** | **75.86** | **86.42** | **40.13** | **74.09** | **85.89** |

, where $\mathcal{L}_{CL}$ is contrastive loss described in the Equation. 1, $R$ is the ranking matrix between two sets of audio and caption, and $\epsilon$ is the coefficient of the entropic regularization term. The experiment shows that entropic regularization can help to boost the performance of contrastive loss from 33.9% to 34.04% and from 39.4% to 44.11% for text-to-audio and audio-to-text retrievals, respectively. However, there is a remarkable gap in performance between the m-LTM and the best performance acquired by contrastive loss with entropic regularization at $\epsilon = 0.05$.

We also provide an additional ablation study for choosing the best hyperparameter $\epsilon$ for the m-LTM framework in Table. 8 and conducted across datasets retrieval experiment to illustrate the robustness of the m-LTM framework compared with triplet and contrastive loss in Table. 9.

**Table 8:** Further ablation study on AudioCaps dataset for choosing the hyperparameter $\epsilon$ for the minibatch learning-to-match framework utilizing Mahalanobis distance.

| Epsilon | Text->Audio | | | Audio->Text | | |
|---|---|---|---|---|---|---|
| | R@1 | R@5 | R@10 | R@1 | R@5 | R@10 |
| 0.01 | 0.10 | 0.52 | 1.04 | 0.10 | 0.31 | 0.41 |
| 0.02 | 0.17 | 0.77 | 2.06 | 0.45 | 1.31 | 3.78 |
| 0.03 | 38.28 | 74.85 | 85.13 | 49.88 | 79.77 | 89.86 |
| 0.04 | 37.22 | 73.58 | 85.37 | 48.38 | 80.04 | 90.17 |
| 0.05 | **39.10** | **74.06** | **85.78** | **49.94** | **80.77** | **90.49** |

**Table 9:** The robustness of minibatch learning-to-match, triplet, and contrastive loss for audio-text retrieval across datasets.

| Train-Test | Method | Text->Audio | | | Audio->Text | | |
|---|---|---|---|---|---|---|---|
| | | R@1 | R@5 | R@10 | R@1 | R@5 | R@10 |
| AudioCaps->Clotho | Triplet | 10.41 | 27.38 | 38.04 | 11.39 | 28.42 | 39.80 |
| | CL | 12.31 | 30.93 | 43.44 | 13.87 | 33.30 | 43.54 |
| | m-LTM | **15.01** | **35.42** | **47.71** | **19.42** | **37.03** | **48.61** |
| Clotho->AudioCaps | Triplet | 14.33 | 40 | 54.42 | 17.34 | 43.05 | 55.69 |
| | CL | 14.64 | 38.64 | 53.58 | 17.76 | 42.11 | 55.59 |
| | m-LTM | **17.38** | **43.78** | **58.87** | **22.04** | **47.85** | **61.65** |

