# OpenReview forum: "Revisiting Deep Audio-Text Retrieval Through the Lens of Transportation"
_ICLR.cc/2024/Conference — ICLR 2024 poster_

### Official Review · Reviewer_Rq97 · 2023-10-25

**Soundness:** 3 good
**Presentation:** 3 good
**Contribution:** 3 good
**Rating:** 8
**Confidence:** 4

**Summary:**

This paper provides a novel method via learning-to-match mechanism, approaching current audio-text retrieval framework and learn the embedding space. The method itself is based on optimal transport, with solid mathematical foundations. The method itself leads to notable  improvements in cross-modal retrieval on common audio datasets, with event detection as the downstream task and additional analysis. Although there are some gaps between the framework design and the scenarios it targets to, it is a well-formed study, which the reviewers learns a lot from.

**Strengths:**

1. The study itself is well-motivated, novel and practical.
2. The mathematical foundation is good.
3. The experiments have been done on commonly-known datasets and the improvements are clearly-observed.

**Weaknesses:**

1. Apart from the performance, it would be good to also show the acquired network architecture, and run-time efficiency for cross-modal inference.
2. The reference of "noise" is not clear and potentially confusing, even with clear references. When talking about the noise, it can be many things. Especially for speech people who are very likely refer to this paper, seems like the definition of "noise" is different from real-world interruption - it is totally fine, but please spend some text on clarifying it in the introduction and methodology (aka Section 3.3).

Also some minor issues:
1. Although I did not find significant grammatical flaws, please do a thorough check on the language usage. For example, at the beginning of Section 4, "Cross-modal matching is an essential task....."sounds a bit weird.

**Questions:**

Most of the questions have been asked as weaknesses in the above section. Please answer them.

I also have additional trivial questions.
1. Do you think if it is possible to open-source the models?
2. Do you think your models will be benefitted by further fine-tuning the network model or pre-trained encoders? Or you think basically that's it (which is totally fine)?
3. Do you think your model can be adopted to other sound-related tasks, with rather little amount of data for domain adaptation? Or maybe collection of experts is needed?
4. Would you classify your approach as essentially an "unsupervised" or "supervised" contrastive loss?
5. Do you see any possibility on developing a "parameter-free" variant of your framework?

**Details Of Ethics Concerns:**

The reviewer does not find any notable concern on the ethical issues.

---

> ### Author Response · Authors · 2023-11-17
> **Response to reviewer Rq97**
>
> We appreciate the valuable time and feedback of reviewer to improve your work. We have revised our manuscript and would like to address your concerns outlined below
>
> **Q1:** Apart from the performance, it would be good also to show the acquired network architecture and run-time efficiency for cross-modal inference.
>
> **A1:** The focus of our work is to learn an effective joint embedding space for audio and text retrieval. We acknowledge the importance of run-time efficiency and leave that to our future work.
>
> **Q2:** The reference of "noise" is not clear and potentially confusing, even with clear references ...
>
> **A2:** We totally agree with your suggestion and have added a sentence in the third paragraph of the introduction to articulate noisy data in the audio-text matching task.
>
> **Q3:** Some minor issues in language usage
>
> **A3:** We have proofread the manuscript and corrected all language usage.
>
> **Q4:** Do you think if it is possible to open-source the models?
>
> **A4:** We will definitely publish our source code and pretrained models to the community.
>
> **Q5:** Do you think your models will be benefitted by further fine-tuning the network model or pre-trained encoders? Or you think basically that's it (which is totally fine)?
>
> **A5:** We leverage pre-trained encoders for our work(also used by prior works) to improve performance and help models converge quickly. Therefore, good pre-trained encoders could help to improve the performance of audio-text retrieval tasks. Regarding further fine-tuning, we think that if we finetune our pre-trained models on a diverse and high-quality dataset, it could enhance the model’s performance like most deep learning models.
>
> **Q6:** Do you think your model can be adopted to other sound-related tasks, with rather little amount of data for domain adaptation? Or maybe collection of experts is needed?
>
> **A6:** Our model could be used for other downstream tasks by finetuning the pretrained model, for example, text-to-audio generation. However, we need an appropriate decoder/adapter model for decoding embedding space to data space.
>
> **Q7:** Would you classify your approach as essentially an "unsupervised" or "supervised" contrastive loss?
>
> **A7:** We would like to classify our approach as a supervised method since the m-LTM needs to use aligned audio-text pairs for training.
>
> **Q8:** Do you see any possibility on developing a "parameter-free" variant of your framework?
>
> **A8:** We assume that you are asking about replacing the Mahalanobis distance with a parameter-free one, like kernel distance. We thought about that but left it for future work.

---

> > ### Comment · Reviewer_Rq97 · 2023-11-21
> >
> > Thanks for addressing all my questions with details! I do not have further questions to ask for now.

---

### Official Review · Reviewer_Mpor · 2023-10-31

**Soundness:** 4 excellent
**Presentation:** 3 good
**Contribution:** 3 good
**Rating:** 6
**Confidence:** 4

**Summary:**

This paper describes an approach for audio-text retrieval using a minibatch version of Learning-to-Match with an optimal transport optimization .  The result of this approach is strong retrieval performance across three datasets.

**Strengths:**

The algorithmic design of this approach is well motivated and (for the most part) well-described.  (The application of Projection Gradient Descent is effective.)

The performance is particularly good compared to triplet and contrastive loss.

**Weaknesses:**

Retrieval applications have an expectation of scaling.  Ideally a single query would be used to retrieve one or more corresponding examples from an extremely large source.  However, in this paper the datasets (particularly the test sets) have a fairly small source to retrieve from (a few thousand examples typically).  It would strengthen the work substantially to demonstrate the capabilities of the algorithm to scale to instances where there are orders of magnitude more examples to retrieve from than queries.

There is a claim that the correspondence in Figure 1 is obviously better in figure 1b than figure 1a.  I think this is not as obvious as claimed.  I don’t think this image adds particularly value  to the understanding of the work.

**Questions:**

Partial Optimal Transport appears to be performed by noising some percentage of the data prior to learning.  When this noise is applied with a batch size of 2, there is a substantial likelihood that both batch elements will contain incorrect (corrupted) examples.  Is there any risk in this from an algorithmic perspective?

How does this approach deal with instances where no match is available?

The authors describe one of the positive attributes of this work as identifying a joint embedding space for both speech and text through which to perform retrieval.  Has this embedding space been used for other downstream tasks? This would strengthen the argument that this is a good joint space rather than solely useful for retrieval.

How effectively does the learned representation work across corpus?  For example, when training on Audiocaps, how effective is retrieval of Clotho test sets and vice versa? This would be a more effective measure of the robustness of the algorithm than the intra-corpus analyses.

The delta_gap described in table 4 is the difference between the means of the embeddings.  First, is this gap an absolute value? it’s remarkable that mean(f(x_i)) is always greater than mean(g(y_i)).  Second, is the corpus mean of text and audio embeddings a reasonable measure of performance? it seems like this measure could be easily gamed by normalizing the embedding spaces to have a 0-mean, but this wouldn’t add anything to the transferability of the embedding.  Also distance in the embedding space isn’t well defined.  It’s not clear that a unit of distance in the triplet loss space can reasonably be compared to a unit in the m-LTM space.

---

> ### Author Response · Authors · 2023-11-18
> **Response to reviewer Mpor (1/2)**
>
> We appreciate the valuable time and feedback of reviewer to improve your work. We have revised our manuscript and would like to address your concerns outlined below
>
> **Q1:** Retrieval applications have an expectation of scaling. Ideally a single query would be used to retrieve one or more corresponding examples from an extremely large source. However, in this paper the datasets (particularly the test sets) have a fairly small source to retrieve from (a few thousand examples typically). It would strengthen the work substantially to demonstrate the capabilities of the algorithm to scale to instances where there are orders of magnitude more examples to retrieve from than queries.
>
> **A1:** We agree that scalability is important for retrieval applications. An accurate similarity function for ranking is also crucial. In this work, we focus on measuring the similarity of examples in a rich and expressive joint embedding space designed for audio-text retrieval problems. We leave the investigation of the scalability issue to future work.
>
> **Q2:** There is a claim that the correspondence in Figure 1 is obviously better in figure 1b than figure 1a. I think this is not as obvious as claimed. I don’t think this image adds particularly value to the understanding of the work.
>
> **A2:**  Figure. 1 demonstrates the joint embedding space for zero-shot sound event detection setting. We would like to show that the joint embedding space learned from our framework is more expressive than contrastive learning. All sound event labels are described in plain text by the template “this is a sound of {sound event}.” Each text embedding in Figure. 1 is equivalent to a sound event label. The sound event detection is now converted to an audio-text retrieval task, therefore, a compact and clustering embedding space is beneficial to the performance. Figure. 1b shows that the m-LTM framework encourages the compact clustering of audio and event texts in the embedding space compared with contrastive loss shown in Figure. 1a. The claim is supported by the performance of zero-shot sound event detection experiment shown in Table. 3.
>
> **Q3:** Partial Optimal Transport appears to be performed by noising some percentage of the data prior to learning. When this noise is applied with a batch size of 2, there is a substantial likelihood that both batch elements will contain incorrect (corrupted) examples. Is there any risk in this from an algorithmic perspective?
>
> **A3:** We agree that there is a risk in this when the batch size is very small. The underlying assumption of our algorithm is that there is at least a proportion of matched examples in a batch. If a batch is too small, either there are no matched examples, or all are matched. Hence, small batches are not the focus of this algorithm.  The intuition of utilizing POT for noise correspondence data is to discard as many noisy pairs as possible. In the case of a small batch size, there is a risk of a whole minibatch of training data being noisy, POT regularization fails to handle that circumstance since it always imposes a percentage of matching(the mass parameter for POT) between two sources. The risk of a corrupted minibatch is critical for all existing methods, including my proposed approach. However, with a large enough minibatch, POT regularization is able to mitigate the harm of misaligned training data.
>
> **Q4:** How does this approach deal with instances where no match is available?
>
> **A4:** The POT approach deals with misaligned instances within a minibatch by a parameter, which forces only to match a percentage of audio-text pairs, the total transportation mass between two sources $0 \leq s \leq 1$. The total transportation mass of POT acts as a regularizer to discard the less certain matches, therefore, it is able to mitigate the harmfulness of mismatched training data.
>
> **Q5:** The authors describe one of the positive attributes of this work as identifying a joint embedding space for both speech and text through which to perform retrieval. Has this embedding space been used for other downstream tasks? This would strengthen the argument that this is a good joint space rather than solely useful for retrieval.
>
> **A5:** We conducted a zero-shot setting for a downstream task, sound event detection, on the ESC-50 test set in the Table. 3 to illustrate the expressiveness of the learned join embedding space.

---

> ### Author Response · Authors · 2023-11-18
> **Response to reviewer Mpor (2/2)**
>
> **Q6:** How effectively does the learned representation work across corpus? For example, when training on Audiocaps, how effective is retrieval of Clotho test sets and vice versa? This would be a more effective measure of the robustness of the algorithm than the intra-corpus analyses.
>
> **A6:** To answer your question, we compare our method with the baselines in the cross-corpus setting. As shown in the table below, the m-LTM is more robust across audio-text retrieval datasets than triplet and contrastive loss. Regarding training on the AudioCaps and testing on the Clotho dataset, our proposed method acquires the highest performance, 15.01%, in terms of R@1 for the text-to-audio retrieval task, compared with 10.41% and 12.31% for triplet and contrastive loss respectively. The same observation is seen for the setting training on Clotho and then testing on AudioCaps dataset.
>
> |     Train-Test    |  Method |       |  T->A |       |       |  A->T |       |
> |:-----------------:|:-------:|:-----:|:-----:|:-----:|:-----:|:-----:|:-----:|
> |                   |         |  R@1  |  R@5  |  R@10 |  R@1  |  R@5  |  R@10 |
> |                   | Triplet | 10.41 | 27.38 | 38.04 | 11.39 | 28.42 | 39.80 |
> | AudioCaps->Clotho |    CL   | 12.31 | 30.93 | 43.44 | 13.87 | 33.30 | 43.54 |
> |                   |  m-LTM  | 15.01 | 35.42 | 47.71 | 19.42 | 37.03 | 48.61 |
> |                   | Triplet | 14.33 |   40  | 54.42 | 17.34 | 43.05 | 55.69 |
> | Clotho->AudioCaps |    CL   | 14.64 | 38.64 | 53.58 | 17.76 | 42.11 | 55.59 |
> |                   |  m-LTM  | 17.38 | 43.78 | 58.87 | 22.04 | 47.85 | 61.65 |
>
> **Q7:** The delta_gap described in table 4 is the difference between the means of the embeddings. First, is this gap an absolute value? it’s remarkable that mean(f(x_i)) is always greater than mean(g(y_i)). Second, is the corpus mean of text and audio embeddings a reasonable measure of performance? it seems like this measure could be easily gamed by normalizing the embedding spaces to have a 0-mean, but this wouldn’t add anything to the transferability of the embedding. Also distance in the embedding space isn’t well defined. It’s not clear that a unit of distance in the triplet loss space can reasonably be compared to a unit in the m-LTM space.
>
> **A7:** The gap value is the length of the discrepancy vector $\vec{\Delta_{gap}} = \frac{1}{n}\sum_{i=1}^n f(x_i) - \frac{1}{n}\sum_{i=1}^n g(y_i)$ which is subtracting vector between the mean of embedding vector of two modalities, we described it in the second paragraph of section 5.1. Therefore, the gap is always positive. The discrepancy between the two modalities proved that it is an effective metric for studying the transferability of a jointed embedding space[1]. We use Euclidean distance on the embedding space to measure the modality metric. To achieve a fair comparison, all embedding is normalized to unit vectors and then used to compute the modality gap.
>
> [1] Mind the Gap: Understanding the Modality Gap in Multi-modal Contrastive Representation Learning.

---

### Official Review · Reviewer_efev · 2023-11-03

**Soundness:** 3 good
**Presentation:** 3 good
**Contribution:** 2 fair
**Rating:** 6
**Confidence:** 4

**Summary:**

The authors revisit the "Learning to Match" (LTM) framework (Li et al., 2018), investigating its utilization to learn cross-modal embeddings for text-to-audio and audio-to-text retrieval.

LTM is based on entropic inverse optimal transport (eq. 3), where the goal is to learn the underlying ground metric c which minimizes the KL divergence of the optimal transfer plan $\pi^{XY}$ determined from (eq. 3) and the empirical joint distribution (eq. 4). Here c is taken as the Mahalanobis distance (eq. 9) between the text and audio embeddings, and a minibatch version (m-LTM) is proposed, where deep networks are used to learn the embeddings, making the parameters of the cost function c the Mahalanobis matrix, and the parameters of the embedding networks.

Results on the AudioCaps and Clotho datasets show large gains over previous approaches (table 1), and large gains over triplet and bi-directional contrastive (eq. 1) losses (table 2). Large gains in terms of zero shot sound event detection (table 3) and modality gap (table 4) are also shown. Large gains in noise tolerance are also shown (table 5). Ablations around using POT for additional robustness show small but consistent gains, and ablations on utlizing Mahalanobis vs. L2 distance also show small but consistent gains.

**Strengths:**

- Well motivated, inverse optimal transport seems worth exploring in the context of deep networks and minibatch training.
- Strong results. The approach consistently outperforms existing SOTA text-audio retrieval results on the most popular datasets, and the most widely used contrastive objectives.
- Generally well presented.

**Weaknesses:**

- As their results are much better than previous approaches and standard contrastive training methods, I feel that this warrants further investigation. The training sets for AudioCaps and Clotho are rather small at 46K and 5K audio examples, respectively, and so regularization may be a very important factor. Their m-LTM approach is entropy regularized, while their Triplet and Constrastive baselines are not. An entropy-regularized constrastive loss baseline is the most natural analog here, and would more firmly establish the importance of the optimal transport formulation. This feels essential to establishing the significance of the method and results.
- The m-LTM method presented is somewhat lower in novelty as a variation on LTM that investigates the use of minibatch training and deep networks, but 'revisiting' is explicitly called out in the title, and this seems a worthy exploration.
- There are a few grammatical errors throughout the paper, but the paper is in general well structured, and adequately well written. A glaring exception is the abstract, which is in really poor shape. Authors, please resolve this.
- In table 6, the best results for the ground metric hyperparameter $\epsilon$ are at 0.05, but no values less than it are tested, while the results at 0.5 are very poor, suggesting that results for $\epsilon<0.05$ should be instead included.

**Questions:**

See previous section.

---

> ### Author Response · Authors · 2023-11-18
> **Response to reviewer efev (1/2)**
>
> We appreciate the valuable time and feedback of reviewer to improve your work. We have revised our manuscript and would like to address your concerns outlined below
>
> **Q1:** A comparison with the entropy regularized contrastive loss baseline
>
> **A1:** Below are the results of the baseline (CL with entropy regularized) suggested by Reviewer efev, in comparison with our method and the one without entropy regularization (CL w/o). More specifically, the baseline is trained by using contrastive loss and entropy regularized on the matching matrix, $L(\theta,\phi) = L_{CL} + \epsilon*L_{ent}(R)$, where $\mathcal{L}_{CL}$ is contrastive loss described in the Equation. 1, R is the ranking matrix between two sets of audio and caption and $\epsilon$ is the coefficient of entropy regularized term. We have reported the experiment in Table. 7 in Appendix A.4. The experimental results demonstrate that entropy regulation can boost audio-text retrieval performance slightly, however, there is a significant gap in performance between the m-LTM and contrastive loss with entropy regularization.
>
> **Q2:** The m-LTM method presented is somewhat lower in novelty as a variation on LTM that investigates the use of minibatch training and deep networks, but 'revisiting' is explicitly called out in the title, and this seems a worthy exploration.
>
> **A2:** We agree that adapting the learning-to-match framework for deep learning is a valuable exploration. However, we acknowledge that the title and abstract can not be modified during the discussion stage. We will definitely change our title regarding your comment when submitting the camera-ready version.
>
> **Q3:** There are a few grammatical errors throughout the paper, but the paper is, in general well structured and adequately well-written. A glaring exception is the abstract, which is in really poor shape. Authors, please resolve this.
>
> **A3:** We have corrected all the grammatical errors and will definitely restructure the abstract when submitting the camera-ready version. Here is our new abstract:
>
> The Learning-to-match(LTM) framework proves to be an effective inverse optimal transport approach for learning the underlying ground metric between two sources of data, facilitating subsequent matching. However, the conventional LTM framework faces scalability challenges, necessitating the use of the entire dataset each time updating parameters of ground metric. In adapting LTM to the deep learning context, we introduce the mini-batch Learning-to-match (m-LTM) framework for audio-text retrieval problems. This framework leverages mini-batch subsampling and Mahalanobis-enhanced family of ground metrics. Moreover, to cope with misaligned training data in practice, we propose a variant using partial optimal transport to mitigate the harm of misaligned data pairs in training data. We conduct extensive experiments on audio-text matching problems using three datasets: AudioCaps, Clotho, and ESC-50. Results demonstrate that our proposed method is capable of learning rich and expressive joint embedding space, which achieves SOTA performance. Beyond this, the proposed m-LTM framework is able to close the modality gap across audio and text embedding, which surpasses both triplet and contrastive loss in the zero-shot sound event detection task on the ESC-50 dataset. Notably, our strategy of employing partial optimal transport with m-LTM demonstrates greater noise tolerance than contrastive loss, especially under varying noise ratios in training data on the AudioCaps dataset.

---

> > ### Author Response · Authors · 2023-11-18
> > **Response to reviewer efev (2/2)**
> >
> > **Q4:** In table 6, the best results for the ground metric hyperparameter $\epsilon$ are at 0.05, but no values less than it are tested, while the results at 0.5 are very poor, suggesting that results for $\epsilon$<0.05 should be included instead.
> >
> > **A4:** $\epsilon$ is a critical hyperparameter for our framework, $\epsilon$ controls the effect of entropic regularization to compute the optimal plan. As mentioned in the ablation study section, if $\epsilon$ is too high, the optimal plan will be a uniform distribution. In the case of $\lim{\epsilon \to 0}$, the optimal plan will converge to the exact optimal transportation plan. We provide additional experiment on $\epsilon$ value lower than 0.05:
> >
> > | Epsilon |    |   T->A    |       |    |  A->T     |       |
> > |:-------:|:-----------:|:-----:|:-----:|:-----------:|:-----:|:-----:|
> > |         |     R@1     |  R@5  |  R@10 |     R@1     |  R@5  |  R@10 |
> > |   0.01  |     0.10    |  0.52 |  1.04 |     0.10    |  0.31 |  0.41 |
> > |   0.02  |     0.17    |  0.77 |  2.06 |     0.45    |  1.31 |  3.78 |
> > |   0.03  |    38.28    | 74.85 | 85.13 |    49.88    | 79.77 | 89.86 |
> > |   0.04  |    37.22    | 73.58 | 85.37 |    48.38    | 80.04 | 90.17 |
> > |   0.05  |    39.10    | 74.06 | 85.78 |    49.94    | 80.77 | 90.49 |
> >
> > As shown in the above table, if the value of $\epsilon$ is lower than 0.02, the performance of audio-text matching plunges from 38.28 to 0.17 and 49.88 to 0.45 in terms of R@1 metric for text-to-audio and audio-to-text retrieval respectively. The performance of the model at $\epsilon=[0.3, 0.4]$ are both lower than the performance of the model at $\epsilon$=0.05. Overall, the above results show that  $\epsilon$=0.05 is indeed the best hyperparameter in our experiments.

---

### Author Response · Authors · 2023-11-18
**General Response and Revised Manuscript Uploaded**

We thank all reviewers for their insightful feedback and suggestions to improve the manuscript. We have updated the manuscript based on the reviewer's comments. The main changes are summarized as follows:

1. We have added a sentence in the third paragraph of the introduction to clarify noisy data for multimodal matching. Specifically, “The noisy data for multimodal matching is misaligned data pairs due to the data collection process”.

2. We have corrected the language usage of the first sentence in Section. 4 and checked thoroughly for language usage.

3. We have added an experiment to compare the contrastive loss with entropic regularization and reported the ablation study for $\epsilon<0.05$ in the appendix(Appendix A.4) according to the reviewer's “efev” suggestion.
4. We have reported the across-corpus performance experiment in the appendix(Appendix A.4) according to the reviewer’s “Mpor” comment.

---

### Comment · Area_Chair_vPiQ · 2023-11-21
**Reminder to reviewers to participate in the author/reviewer discussion**

Dear reviewers, this is a reminder that the author/reviewer discussion period ends November 22.

This discussion is indeed supposed to be a dialog, so please respond to
the comments from the authors as well as the updated manuscript.

@Reviewer Rq97 - thank you for having already done this!

AC

---

### Meta-Review · Area_Chair_vPiQ · 2023-12-05

**Metareview:**

## Scientific Claims and Findings
The paper proposes a method for learning text and audio encoders and a joint embedding space using a minibatch variant of learning to match, which is based on entropically regularized optimal transport, and a Mahalanobis distance ground metric. The paper also proposes a robust variant of the algorithm that employs partial optimal transport to cope with label noise in the training data. Audio-text retrieval experiments on the AudioCaps and Clotho datasets show that the proposed algorithm outperforms competing, state-of-the-art baselines; that the minibatch learning-to-match loss yields better performance than either the triplet loss or contrastive loss, including an entropically regularized variant of the contrastive loss; and that the partial optimal transport approach is more robust to label noise. Zero-shot sound event detection tests on ESC50 show that the embedding space learned with the proposed minibatch learning-to-match loss transfers to other downstream tasks better than spaces learned using either the triplet or contrastive loss, and measurements of the modality gap on AudioCaps, Clotho, and ESC50 show that it is much smaller for the proposed loss. Ablation experiments illustrate that the Mahalanobis distance provides improved performance compared to Euclidean distance, illustrate the importance of proper selection of the entropic regularization weight in the optimal transport loss, and show that it is important to match the quantity of excluded mass in the partial optimal transport procedure to the expected amount of label noise in the training data.

## Strengths
- Nice application of learning-to-match, and the minibatch variant of the algorithm is a useful contribution.
- Well-designed and convincing empirical evaluation of the proposed algorithm.

## Weaknesses
- The scalability of the approach is not really discussed in the paper.
- The paper is still not making correct use of the \citep and \citet commands. The authors need to fix this problem to make the paper more readable. The LaTeX template "iclr2023_conference.tex" distributed in the paper kit actually discusses the use of these commands.

**The next three weaknesses were called out by one reviewer, but were provided too late for the authors to respond to them. If at all possible, the authors should attempt to address these issues in the final version of the paper.**
- The details regarding the hyperparmeter search over the temperature of the baseline contrastive objective (eq. 1) are not in the paper. Considering that the corresponding grounding metric temperature parameter $\epsilon$  was crucial to performance, details should be provided.
- The whitening of the marginal priors that is done via the Sinkhorn iteration to arrive at $\pi^{XY}$ is also done at test time (eq. 5), which is a form of test-time normalization/adaptation which utilizes knowledge of all test queries. The baselines, in constrast, do no test-time normalization/adaptation. Some discussion around and analysis ablating the effects of this test-time normalization should be added to the paper (i.e. baseline results with this normalization at test time, and the proposed approach without this normalization at test time).
- Details regarding how the Sinkhorn iteration is applied at test time should also be included. Computing the optimal plan requires knowledge of all of the test queries, which is rarely the case in practice. This seems like a significant limitation of the approach, and warrants discussion.

**Justification For Why Not Higher Score:**

The question of scalability remains unaddressed.

**Justification For Why Not Lower Score:**

The paper has a solid algorithmic contribution that is evaluated in an experimentally sound manner. It's a good contribution, and the paper will likely be useful to other researchers.

---

### Decision · Program_Chairs · 2024-01-16

Accept (poster)